# Line-Edge Roughness from Extreme Ultraviolet Lithography to Fin-Field-Effect-Transistor: Computational Study

**DOI:** 10.3390/mi12121493

**Published:** 2021-11-30

**Authors:** Sang-Kon Kim

**Affiliations:** The Faculty of Liberal Arts, Hongik University, Seoul 04066, Korea; sangkona@hongik.ac.kr

**Keywords:** lithography, lithography simulation, extreme ultraviolet, EUV, line-edge roughness, LER, stochastic simulation, fin-field-effect-transistor, FinFET

## Abstract

Although extreme ultraviolet lithography (EUVL) has potential to enable 5-nm half-pitch resolution in semiconductor manufacturing, it faces a number of persistent challenges. Line-edge roughness (LER) is one of critical issues that significantly affect critical dimension (CD) and device performance because LER does not scale along with feature size. For LER creation and impacts, better understanding of EUVL process mechanism and LER impacts on fin-field-effect-transistors (FinFETs) performance is important for the development of new resist materials and transistor structure. In this paper, for causes of LER, a modeling of EUVL processes with 5-nm pattern performance was introduced using Monte Carlo method by describing the stochastic fluctuation of exposure due to photon-shot noise and resist blur. LER impacts on FinFET performance were investigated using a compact device method. Electric potential and drain current with fin-width roughness (FWR) based on LER and line-width roughness (LWR) were fluctuated regularly and quantized as performance degradation of FinFETs.

## 1. Introduction

According to a scaling-down process, extreme ultraviolet lithography (EUVL) with 13.5-nm wavelength provides a solution to avoid the complex multi-patterning integration and cost [1]. Fin-field-effect-transistor (FinFET) is one of mainstream devices for the post-planar complementary metal-oxide semiconductor (CMOS) because of its efficient suppression of short-channel effect and leakage current [2,3]. However, for 5-nm pattern formation, EUVL has faced many technical challenges towards this paradigm shift to its wavelength platform [4,5]. There are well-known fundamental trade-off relationships among resolution (R), line-edge roughness (LER), and sensitivity (S) that hamper their simultaneous enhancement in chemically amplified resists (CARs) [6]. Hence, LER is one of current challenges limiting EUV applications. LER affects feature size and device malfunctions so significantly that LER reduction with nanometer accuracy is required [7,8,9]. LER and line-width roughness (LWR) are caused by EUV stochastic events such as shot noise of incident photons, chemical concentration shot noise, and molecule reaction-diffusion in resists [10]. Since numbers of photons in EUVL are 14 times smaller than those of ArF lithography, stochastic EUV photons can result in photon shot noise, which makes poor performance in EUV resist [11,12]. In addition, EUV photons contribute to fluctuation of acid generation and reaction with quenchers due to random location of PAG and quencher [13,14]. Therefore, during EUVL processes, stochastic EUV photons initiate random physical and chemical events in terms of multi-photon effect in frequency distribution and cascade and cluster of correlated reactions in special distribution [15]. Phenomena of these stochastic events are LER, LWR, and stochastic defects such as pinching and bridges [16]. Compared to previous LER modeling of ArF resists, precise EUVL process modeling of LER has been a hot issue [17,18,19,20]. The fin and gate critical dimension (CD) LERs of FinFET devices can seriously degrade performance and yield [21,22]. In this sense, it is required to understand LER creation mechanism in EUVL and FinFET performance degradation due to LER [23,24]. Although many studies have determined LER effects of EUVL and FinFETs [25,26], this paper deals with LER effects of EUVL simulation parameters for 5-nm pattern formation and FinFET performance with 5-nm gate length, totally describing LER effects from EUVL processes to FinFET devices. For LER creation in EUVL and LER impacts on FinFET performance, a LER modeling in EUVL processes and an analytical method for FinFET degradation due to LER are described, respectively.

## 2. Simulation Method

### 2.1. LER Modeling in EUVL Processes

Figure 1 shows schematic representation of exposure process. In exposure process in Figure 1, EUV photons are absorbed at random positions of an EUV resist due to exposure parameters such as local light intensity related to imaging system and resist absorption. Incident EUV light collides with an atom of an EUV resist and the atom emits photoelectrons in arbitrary directions through ionization process [13,15]. Each photoelectron moves and stops until its energy becomes lower than a certain threshold energy. Through collisions between a photoelectron and an atom in resist materials, a photoelectron’s direction is changed according to elastic scattering and inelastic scattering. Monte Carlo method can be used to compute a possible set of scattering events for a photoelectron as it travels inside a resist [27,28,29]. By repeating this process many times, a statistically valid and detailed picture of interaction processes can be constructed. For elastic scattering between a photoelectron (or a secondary electron (SE)) and an atom, momentum and kinetic energy of a photoelectron (or a SE) are conserved. The scattering cross-section ($\sigma_{T}$) of the tabulated Mott data for a low energy is
(1)$$\sigma_{T}=\frac{3.0\times 10^{-18}Z^{-1.7}}{E+0.005Z^{1.7}E^{0.5}+0.0007Z^{2}/E^{0.5}}\;,$$
where E and Z are incident energy and atomic number, respectively. For inelastic scattering between a photoelectron and an atom’s electron, this scattering not only changes a photoelectron’s direction, but changes a photoelectron’s energy. For outer shell excitation, Moller cross-section with energy limit is
(2)$$\sigma_{T}=\frac{\pi e^{4}}{E^{2}}\left\{\frac{1}{\varepsilon_{c}}-\frac{1}{1-\varepsilon_{c}}-\ln\left(\frac{\varepsilon_{c}}{1-\varepsilon_{c}}\right)\right\}\;,$$
where $\varepsilon_{c}$ is a coefficient of transferred minimum energy. For inner shell ionization with binding energy, Vriens cross-section is
(3)$$\sigma_{T}=\frac{\pi e^{4}}{E^{2}\left(1+2U_{i}\right)}\left\{\frac{5}{3U_{i}}-1-\frac{2}{3}U_{i}+\Phi \frac{\ln U_{i}}{1+U_{i}}\right\}\;,\;\Phi =\cos\left[-{\left(\frac{R_{y}}{1+U_{i}}\right)}^{1/2}\ln U_{i}\right]\;,$$
where $R_{y}$ and $U_{i}$ are Rydberg energy and binding energy normalized by the primary energy, respectively. A portion of the absorbed energy is transmitted to the atom’s electron, and the atom becomes excited or ionized. Incident photoelectrons (or SEs) produce SEs by outer shell electron excitation. Moller cross-section is used for this excitation with free electrons [30]. CSDA (continuously slowing down approximation) model is used as an energy loss model in Bethe equation:(4)$${\left[\frac{dE}{ds}\right]}_{cont}={\left[\frac{dE}{ds}\right]}_{Bethe}-{\left[\frac{dE}{ds}\right]}_{dis},\;{\left[\frac{dE}{ds}\right]}_{Bethe}=\frac{2\pi e^{4}}{E}Z\ln\left(\frac{1.166E}{J^{\prime }}\right)\;,$$
(5)$$\begin{array}{ll} {\left[\frac{dE}{ds}\right]}_{dis}= & \frac{{\pi e}^{4}N}{E}[\sum^{\;}\frac{Z_{i}}{1+2U_{i}}\left\{\frac{2}{3}-3\ln2+\frac{1}{3}\left(1+U_{i}\right)\left(1-2U_{i}\right)+3\ln\left(1+U_{i}\right)-\ln U_{i}\right\}+\\ & Z_{f}\left\{2-3\ln2-\frac{1}{1-\varepsilon_{c}}-2\ln\left(1-\varepsilon_{c}\right)-\ln\varepsilon_{c}\right\}],\;J^{\prime }=\frac{J}{1+kJ/E}\;, \end{array}$$
where s, J, and k are path length along the trajectory, the mean ionization potential of materials, and a value depended on materials, respectively. After spin coating, photo-acid generators (PAGs) is distributed at random positions uniformly inside a chemical amplified resist (CAR). PAGs capture some of SEs and generate photoacids within SE blur range [31]. Therefore, acid generation (AG) rate is corresponding to capture rate, which is depended on probability of SE existence at a PAG site. As a good approximation to this result, a point-spread function ($P_{AG}$) of AG probability with electrons is suggested as
(6)$$P_{AG\left(ionization\right)}=\frac{\phi_{polymer}\int_{0}^{\infty }R_{AG}C_{AG}wd\left(D_{e}t\right)}{\int_{0}^{\infty }w_{t=0}r^{2}dr}\;,$$
where the initial distribution function of thermalized electrons is $4\pi w_{t=0}r^{2}dr=\left(1/r_{0}\right)\exp\left(-r/r_{0}\right)dr$, w is probability density of electrons, $r_{0}$ represents the mean initial separation distance between a thermalized electron and its parent radical cation, and $\phi_{polymer}$ is deprotonation efficiency of polymer radical cations [32,33]. For electron migration after thermalization, equation of low-energy (thermalized) electrons to AGs is
(7)$$\frac{\partial w}{\partial D_{e}t}=\frac{\partial^{2}w}{\partial r^{2}}+\left(\frac{2}{r}+\frac{SN_{+}e}{4\pi \varepsilon_{0}\varepsilon k_{B}Tr^{2}}\right)\frac{\partial w}{\partial r}-4\pi R_{AG}C_{AG}w\;,$$
where $D_{e}$, $k_{e}$, T, $R_{AG}$, $C_{AG}$, e, $\varepsilon_{0}$, ε, and $N_{+}$ represent diffusion constant of electrons, Boltzmann constant, absolute temperature, effective reaction radius, concentration of AGs, elementary electric charge, dielectric constant in vacuum, relative dielectric constant of a resist film, and average number of positive charges, respectively.

For post-exposure bake (PEB) process, acid distribution ($C_{acid}$) in CAR resists catalyzes a thermally-induced reaction with quenches:(8)$$\frac{\partial C_{acid}}{\partial t}=D_{acid}\nabla^{2}C_{acid}-k_{q}C_{acid}C_{q}-k_{loss}C_{acid}\;,$$
(9)$$\frac{\partial C_{q}}{\partial t}=D_{q}\nabla^{2}C_{q}-k_{q}C_{acid}C_{q}\;,$$
(10)$$\frac{\partial C_{p}}{\partial t}=-k_{p}C_{acid}C_{q}\;,$$
where $D_{acid}\left(=k_{0acid}\exp\left(-E_{aacid}/RT\right)\right),$ $D_{q}\left(=k_{0q}\exp\left(-E_{aq}/RT\right)\right),$ $k_{q}\left(=k_{0q}\exp\left(-E_{aq}/RT\right)\right),$ $k_{loss}\left(=k_{0loss}\exp\left(-E_{aloss}/RT\right)\right)$, $k_{p}\left(=k_{0p}\exp\left(-E_{ap}/RT\right)\right),$ $C_{q},$ $C_{p},$ $E_{aacid}$, $E_{aq}$, $E_{ap},$ R, t, and T are diffusion constant of acid, diffusion constant of base quenchers, rate constants of neutralization, rate constant of acid loss, rate constant of deprotection, concentration of base quenchers, concentration of protected unit, activation energy of acids, activation energy of quenches, activation energy of deprotection, ideal gas constant, time, and temperature, respectively [13,33]. For PEB process, equations (8–10) of acid and base quencher diffusion, deprotection reaction, and neutralization can reproduce experimental results [34,35]. When $f_{LER}$ is proportionality constant and m is normalized protected unit concentration, $LER\;\left(\approx f_{LER}/\left(dm/dx\right)\right)$ is proportional to protected unit fluctuation [36,37]. For LER reduction, enhancement of chemical gradient (dm/dx) at boundaries between lines and spaces can be increased through absorption coefficient increase of resist polymer, quantum efficiency of acid generation, effective reaction radius for deprotection, and increase of PAG concentration.

A stochastic model proves useful for prediction of LER without quencher and photon shot noise:(11)$$LER\propto \frac{1}{1-e^{-\pi^{2}\sigma_{D}^{2}/2L^{2}}}\sqrt{1-\left(K_{amp}t_{PEB}\right)\langle m^{*}\rangle \ln\langle m^{*}\rangle {\left(\frac{\sqrt{2}a}{\sigma_{D}}\right)}^{2}\frac{\langle n_{0-block}\rangle }{\langle n_{0-PAG}\rangle }}\;,$$
where $K_{amp}$ is amplification rate constant, $t_{PEB}$ is PEB time, $\langle m^{*}\rangle$ is means value of blocked polymer concentration after PEB, $\sigma_{D}/a$ is a ratio of acid diffusion length to capture range of deblocking reaction, $\langle n_{0-PAG}\rangle$ is mean initial number of PAGs in control volume at exposure start, and $\langle n_{0-blocked}\rangle$ is mean initial number of blocked polymer groups in volume before PEB [38].

### 2.2. LER and LWR Modeling of FinFET

During lithography processes, LER and LWR are factors of EUV stochastic events such as shot noise of incident photons. TCAD has been used to apply LER and LWR to device performance and I-V characteristics [39,40,41,42,43]. Figure 2 describes a FinFET structure. For electric potentials, governing equations of short-channel FinFET in a subthreshold region (low-gate voltage) are
(12)$$\frac{\partial^{2}\varphi_{0}\left(x,\;y\right)}{\partial x^{2}}+\frac{\partial^{2}\varphi_{0}\left(x,\;y\right)}{\partial y^{2}}=\frac{qN_{a}}{\varepsilon_{si}}\;,\;\frac{\partial^{2}\varphi_{1}\left(x,\;y\right)}{\partial x^{2}}+\frac{\partial^{2}\varphi_{1}\left(x,\;y\right)}{\partial y^{2}}=0\;,$$
where $\varphi_{0}$(or $\varphi_{1}$), $N_{a}$, q, and $\varepsilon_{si}$ are zeroth (or first) order of electric potential, doping concentration, electric charge, and silicon permittivity, respectively [44,45]. Using boundary conditions, electric potential ($\varphi_{0}$) without LER can be approximated as a parabolic form:(13)$$\varphi_{0}=C_{0}\left(y\right)+C_{1}x+C_{2}x^{2}\;,$$
(14)$$C_{0}\left(y\right)=V_{SL}+\frac{\left(V_{bi}-V_{SL}\right)\sin h\left(\frac{L-y}{\lambda }\right)}{\sin h\left(\frac{L}{\lambda }\right)}+\frac{\left(V_{bi}+V_{ds}-V_{SL}\right)\sin h\left(\frac{y}{\lambda }\right)}{\sin h\left(\frac{L}{\lambda }\right)}\;,$$
(15)$$C_{1}=0\;,\;\;\;C_{2}=\frac{V_{g}-V_{fb}-C_{0}}{\frac{t_{i}\varepsilon_{si}t_{si}}{\varepsilon_{i}}+\frac{t_{si}^{2}}{4}}\;,$$
where $\lambda \left(=\sqrt{1/2\left(t_{i}\varepsilon_{si}t_{si}/\varepsilon_{i}+t_{si}^{2}/4\right)}\right)$,$\;V_{bi}\left(=E_{g}/\left(2q\right)+kT/q\ln\left(N_{a}/n_{i}\right)\right)$, $V_{SL}\left(=V_{g}-V_{fb}-\left(qN_{a}/\varepsilon_{si}\right)\lambda^{2}\right)$, $V_{fb}$, $E_{g}$, $n_{i}$, L, $t_{i}$, $t_{si}$, $\varepsilon_{i}$, $V_{ds}$, k, and T are a parameter, built-in potential at source end, center potential for a long-channel transistor, flat-band voltage, silicon bandgap energy, intrinsic carrier concentration, channel length, oxide thickness, fin width, oxide permittivity, drain-source voltage, Boltzmann constant, and temperature, respectively. Electric potential ($\varphi_{1}$) with LER can be approximated as a parabolic form:(16)$$\varphi_{1}=\sum_{k}^{\infty }\left[a_{k}\sin h\left(\frac{\pi k}{L}\left(x-\frac{t_{si}}{2}\right)\right)+b_{k}\sin h\left(\frac{\pi k}{L}\left(x+\frac{t_{si}}{2}\right)\right)\right]\sin\left(\frac{\pi k}{L}y\right)\;,$$
(17)$$a_{k}=-\frac{2}{L}\int_{0}^{L}t_{2}C_{2}t_{si}\sin\left(\frac{\pi k}{L}y\right)/\sin h\left(\frac{\pi kt_{si}}{L}\right)dy\;,$$
(18)$$b_{k}=-\frac{2}{L}\int_{0}^{L}t_{1}C_{2}t_{si}\sin\left(\frac{\pi k}{L}y\right)/\sin h\left(\frac{\pi kt_{si}}{L}\right)dy\;,$$
where $t_{1}$ and $t_{2}$ are functions of fin-width roughness (FWR). Drain current ($I_{ds}$) can be described as
(19)$$\begin{array}{ll} I_{ds}dy & =\mu qWQ_{inv}dV=\mu qW\left[\frac{n_{i}^{2}}{N_{a}}e^{-\frac{q}{kT}V}\int_{-t_{si}/2+t_{2}}^{t_{si}/2+t_{1}}e^{\frac{q}{kT}\left(\varphi -V\right)}dx\right]dV\;,\\ & \approx \mu qW\frac{n_{i}^{2}}{N_{a}}e^{-\frac{q}{kT}V}\int_{-t_{si}/2}^{t_{si}/2}e^{\frac{q}{kT}\varphi_{0}}\left(1+\Delta \right)dxdV\;,\; \end{array}$$
where $\Delta =\left(q/\left(kT\right)\right)\varphi_{1}+\left(1/2\right){\left(q\varphi_{1}/\left(kT\right)\right)}^{2}+\cdots$, μ is low-field and temperature-dependent mobility, $Q_{inv}$ is inversion charge density, W is total effective fin-width, and V is quansi-Fermi potential [46,47]. Drain currents $I_{ds0}$ (and $\Delta I_{ds}$) without (and with) LER are, respectively,
(20)$$I_{ds0}=\frac{q\mu W\frac{kT}{q}\frac{n_{i}^{2}}{N_{a}}\left[1-\exp\left(-\frac{V_{ds}}{kT/q}\right)\right]}{\int_{0}^{L}dy/\int_{-t_{si}/2}^{t_{si}/2}e^{\frac{q}{kT}\varphi_{0}}dx}\;,\;\Delta I_{ds}=\frac{q\mu W\frac{kT}{q}\frac{n_{i}^{2}}{N_{a}}\left[1-\exp\left(-\frac{V_{ds}}{kT/q}\right)\right]}{\int_{0}^{L}dy/\int_{-t_{si}/2}^{t_{si}/2}\Delta e^{\frac{q}{kT}\varphi_{0}}dx}\;.\;$$

These theoretical equations were verified with experimental results and simulation results of commercial TCADs in [46,47].

## 3. Results and Discussion

Figure 3 shows Monte Carlo simulations of a photoelectron and secondary electron trajectories by using a hybrid model with Equations (1)–(5) through elastic and inelastic scatterings in Figure 1. Simulation conditions were wavelength (λ = 13.5-nm), incidence angle (θ = 6 deg.), numerical aperture (NA = 0.33), a dipole illumination, resist thickness (20-nm), incident dose of 15 mJ/cm^2^ (10.2 photons/nm^2^), a PHS (C_8_H_8_O)-CAR, and threshold energy ($E_{th}$ = 21 eV).

Figure 4a shows AG probability at EUV absorption points in Equation (6). Monte Carlo method tracked electron trajectories generated by 11 EUV photons. Figure 4b shows numerical simulation of electron migration in Equation (7) by using the forward time and centered space (FTCS):(21)$$\frac{w_{i,j}^{n+1}-w_{i,j}^{n}}{\Delta D_{e}t}=\left(\frac{w_{i+1,j}^{n}-2w_{i,j}^{n}+w_{i-1,j}^{n}}{\Delta r^{2}}\right)+\left(\frac{2}{i\Delta r+r_{i}}+\frac{SN_{+}e^{2}}{4\pi \varepsilon_{0}k_{B}T{\left(i\Delta r+r_{i}\right)}^{2}}\right)\left(\frac{w_{i+1,j}^{n}-w_{i,j}^{n}}{\Delta r}\right)-4\pi R_{AG}C_{AG}w_{j}^{n}.$$

For larger simulation time, probability density of electrons moved to the left more in Figure 4a. Simulation conditions were average number of positive charges ($N_{+}$ = 4.2), shielding effect (S = 0.67), effective reaction radius ($R_{AG}$ = 2.4-nm), AGs concentration ($C_{AG}$ = 10 wt%), relative dielectric constant of resist film (ε = 4), mean initial separation distance ($r_{0}$ = 4-nm), $k_{B}T$ = 4.11 × 10^−21^ J, and diffusion constant of electrons ($D_{e}$ = 1.0 nm^2^ s^−1^). Therefore, LER formation was caused by initial acid distribution due to fluctuation of acid concentration at image boundary.

Figure 5 shows numerical simulation of a negative CAR without quenchers in Equations (8) and (10) by using FTCS:(22)$$\frac{C_{acid}{}_{i,j}^{n+1}-C_{acid}{}_{i,j}^{n}}{\Delta t}=-k_{loss}C_{acid}{}_{j}^{n}+D_{acid}\left(\frac{C_{acid}{}_{i+1,j}^{n}-2C_{acid}{}_{i,j}^{n}+C_{acid}{}_{i-1,j}^{n}}{\Delta x^{2}}+\frac{C_{acid}{}_{i,j+1}^{n}-2C_{acid}{}_{i,j}^{n}+C_{acid}{}_{i,j-1}^{n}}{\Delta y^{2}}\right)\;,$$
(23)$$\left[C_{p}\right]={\left[C_{p}\right]}_{t=0}e^{-k_{p}C_{acid}}.$$

For PEB process, concentration of cross-linked polymer was diffused more due to a larger diffusion length in Figure 5. Simulation conditions were rate constant of deprotection ($k_{p}=2.5$) and rate constant of acid loss ($k_{loss}=2.3\times 10^{5}$).

Figure 6a shows $\mathrm{LER}\;\left(\approx f_{LER}/\left(dm/dx\right)\right)$ behaviors due to exposure dose. As exposure dose increased, LER dropped down and then became saturated because exposure fluctuation decreased fast. LER conditions were proportionality constant ($f_{LER}=0.3$), diffusion constant of acid ($D_{acid}=1\;{\mathrm{nm}}^{2}\;s^{-1}$), and diffusion constant of quencher ($D_{q}=1\;{\mathrm{nm}}^{2}\;s^{-1}$). Figure 6b shows trend of LER versus acid diffusion with different values of deprotection capture range, a = 1.0 and 1.5-nm for 5-nm feature. In each case, there was a diffusion length that minimized LER. Below the optimum diffusion length, increasing diffusion improved LER because LER was limited by statistical variance of blocked polymer concentration. However, above the optimum diffusion length, it further increase degraded gradient and worsened the LER, because LER was limited by gradient.

For fluctuation of electric potentials due to FWR four-types in Figure 7a using Equations (16)–(18), although fluctuation ranges were different, graphs of electric potentials $\varphi_{1}\left(0,\;y\right)$ at fat-fin type, thin-fin type, big-source type, and big-drain type were right shift, left shift, down shift, and upper shift, respectively. Figure 7b shows electric potentials $\varphi_{1}\left(x,\;L/2\right)$ of x-distance at 5-nm gate length for a FinFET with FWRs. Sequence of lager fluctuations of electric potential $\varphi_{1}\left(x,\;L/2\right)$ with x-distance along fin width was big-drain type < fat-fin type = thin-fin type < big-source type.

For gate length L = 5-nm, Figure 8 shows absolute drain currents $\left|\Delta I_{ds}\right|$ with FWRs due to gate voltages ($V_{g}$) using Equation (20). Fluctuations of absolute drain currents ($\left|\Delta I_{ds}\right|$) with fat-fin, thin-fin, and big-drain FWRs shifted righter, respectively. However, values of absolute drain currents ($\left|\Delta I_{ds}\right|$) increased in terms of larger gate length. Larger currents in Figure 8 can be considered as limit of simple FinFET performance with gate lengths L = 5-nm. Simulation conditions were drain-source voltage ($V_{ds}$ = 0.05 V), intrinsic carrier concentration ($n_{i}$ = 1.5 × 10^10^ cm^−3^), doping concentration ($N_{a}$ = 10^17^ cm^−3^), channel length (L = 5-nm), oxide thickness ($t_{i}$ = 0.72-nm), fin width ($t_{si}$ = 5-nm), oxide permittivity ($\varepsilon_{i}$ = 3.9$\varepsilon_{0}$), permittivity ($\varepsilon_{0}$ = 8.854 × 10^−12^ C^2^N^−1^m^−2^), Boltzmann constant and temperature (kT = 0.026 eV), electric charge *(*q = 1.6 × 10^−19^ C), gate voltages ($V_{g}$ = 0.2 V), total effective fin-width (W = 10-nm), low-field and temperature-dependent mobility (μ = 100 cm^2^V^−1^s^−1^), and amplitude of function FWRs (A = 1.0 × 10^−9^ m).

A neural network such as Taguchi method is a powerful method for integration of design of experiments (DOE) with parametric optimization of processes, yielding desired results by using an orthogonal array experiments that provide much-reduced variance for experiments. Hence, this method is a simple and efficient method to find best range of designs for quality, performance, and computational cost by using a statistical measure of performance called signal-to-noise ratio (S/N). S/N ratio is defined as mean (signal) ratio to standard deviation (noise). S/N ratios are lower-the-better (LB), higher-the-better (HB), and nominal-the best (NB). S/N ratio for LER is LB (lower-the-better) criterion:(24)$$S/N=-10\log\left(\frac{1}{n}\sum^{\;}y^{2}\right)\;,$$
where y is observed data and n is number of observations. Parameter level combination that maximizes appropriate S/N ratio is optimal setting [48,49].

Figure 9a shows sensitivity of EUVL parameters on LER by using Taguchi method in minitab^TM^, a commercial tool. According to a neural method, PEB temperature ($T_{PEB}$) and PEB time ($t_{PEB}$) are dominant factors. This means that PEB process is the most dominant process for LER in EUVL processes. Thus, controlling PEB time is effective in managing LER in experimental processes. Figure 9b shows sensitivity of FinFET parameters on electric potentials and current drains with FWR. According to S/N effects, gate voltage ($V_{g}$) and channel length (L) are more dominant factors for electric potential $\varphi_{1}\left(0,L/2\right)$ and drain current $\Delta I_{ds}$ with FWR. Sensitivity of FWR amplitude on electric potentials and drain currents is similar to sensitivity of oxide thickness. When particle dimension of semiconductors approached near to or below Bohr exciton radius of bulk semiconductor, current performance can be affected by quantum confinement effects [50]. Quantum confinement effects should be considered when modelling of 5-nm FinFET devices.

## 4. Conclusions

An EUVL modeling and a compact device method described LER impacts on 5-nm patterns and FinFET performance with 5-nm gate length, respectively. For EUVL processes, a Monte-Carlo method and a point-spread function were used for scattering events of EUV photons and acid distribution of a CAR, respectively. This simulator successfully performed LER for 5-nm patterns. According to a compact device method, for y-direction along the gate length, electric potentials of 5-nm gate length with fat-fin, thin-fin, big-source, and big-drain FWRs were right shift, left shift, down shift, and upper shift, respectively. For x-direction along fin width due to gate length, sequence of lager fluctuations in electric potentials was big-drain type < fat-fin type = thin-fin type < big-source type. For drain currents with FWRs due to gate voltages, absolute drain currents with fat-fin, thin-fin, and big-drain FWRs shifted righter, respectively. However, larger currents can be caused by limit of the simple FinFET performance. According to a neural network for LER, PEB temperature and PEB time are dominant factors. Gate voltage and channel length are dominant for sensitivity of electric potential and drain current in a FinFET device with FWRs. Therefore, for reduction of LER and FWR effects, values of those dominant parameters should be reduced.

## Figures and Tables

**Figure 1 micromachines-12-01493-f001:**
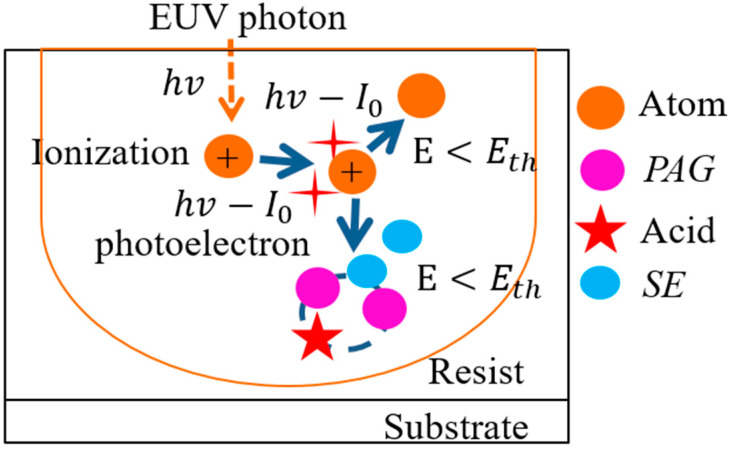
Exposure simulation of Monte Carlo method: molecular processes in a resist-substrate system during EUV exposure. hv, $I_{e}$, E, $E_{th}$, *PAG*, and *SE* are EUV energy, ionization energy of polymers, energy, threshold energy, photo-acid generator, and secondary electron, respectively.

**Figure 2 micromachines-12-01493-f002:**
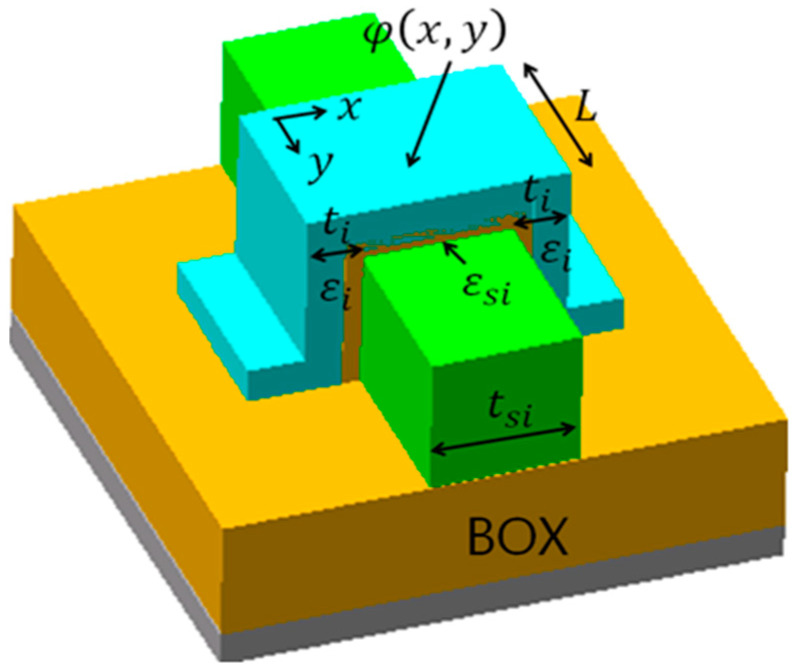
Schematic description of a 3D FinFET device.

**Figure 3 micromachines-12-01493-f003:**
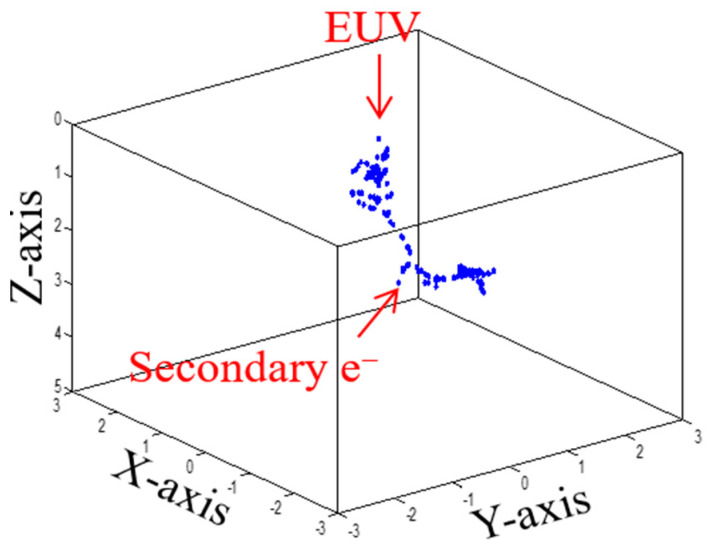
Exposure simulation of Monte Carlo method: a photoelectron and secondary electron trajectories with EUV energy (92.5 eV) due to elastic and inelastic scattering (collision) of a photoelectron with an atom in a polyhydroxystyrene (PHS)-based chemically amplified resist.

**Figure 4 micromachines-12-01493-f004:**
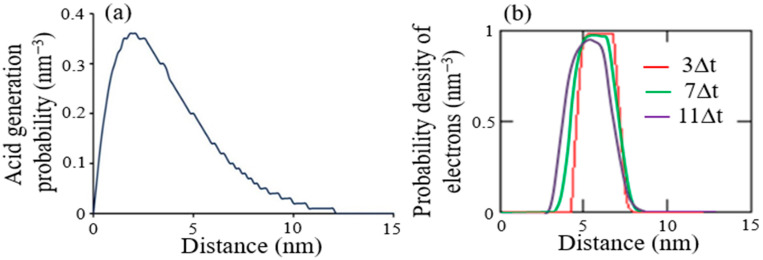
Simulation results: (**a**) probability of acid generation for CAR by Monte Carlo method; (**b**) migration of electron probability density after thermalization in Equation (7). Δt is an arbitrary time interval.

**Figure 5 micromachines-12-01493-f005:**
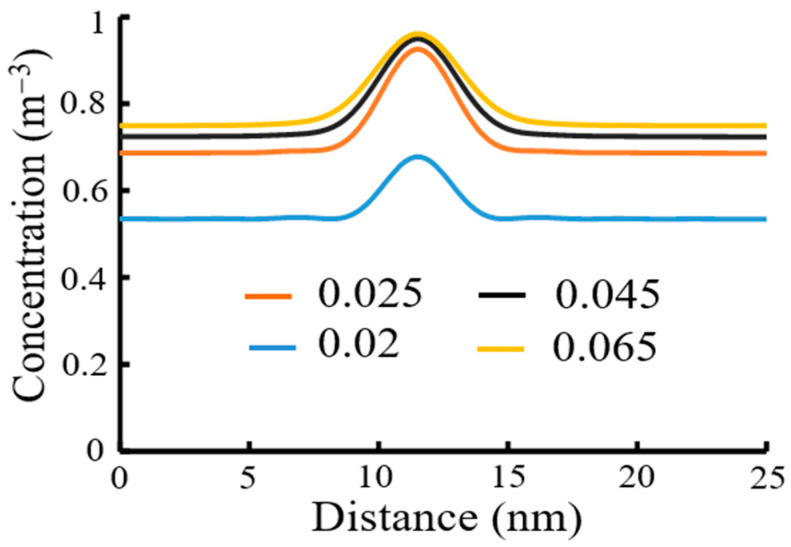
Simulation results for cross-linked polymer concentration of a negative CAR after PEB by using FTCS due to diffusion lengths (*D_acid_*): 0.02 μm; 0.025 μm; 0.045 μm; 0.065 μm.

**Figure 6 micromachines-12-01493-f006:**
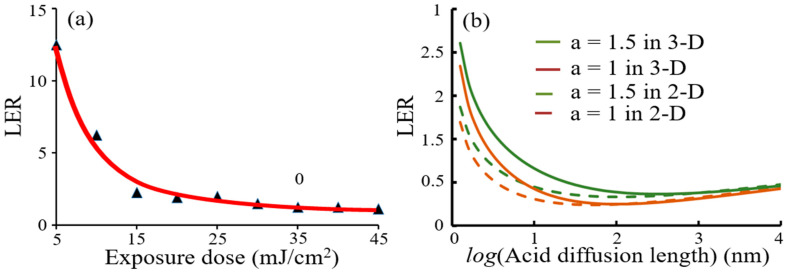
LER simulation results: (**a**) LER dependency of exposure dose; (**b**) prediction of LER trends for 5-nm patterns using two values of the deblocking reaction capture range, a = 1.0-nm and 1.5-nm in 2-Dimension and 3-Dimension calculations.

**Figure 7 micromachines-12-01493-f007:**
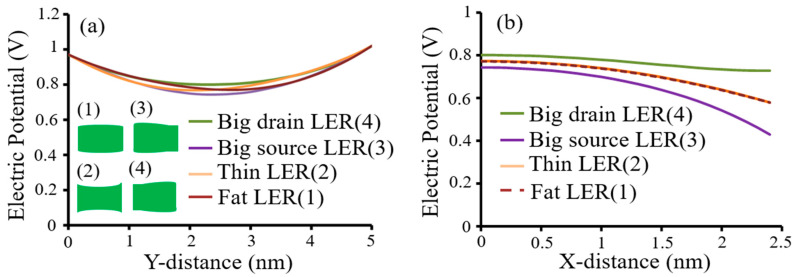
Analytical results: (**a**) electric potentials φ_1_(0, *y*) for a FinFET with FWRs; (**b**) electric potentials $\varphi_{1}\left(x,\;L/2\right)$ of x-distance for a FinFET with FWRs. FWR functions are $t_{1}=A\sin\left(2\pi y/L\right)$ and $t_{2}=-A\sin\left(2\pi y/L\right)$ for fat-fin type, $t_{1}=-A\sin\left(2\pi y/L\right)$ and $t_{2}=A\sin\left(2\pi y/L\right)$ for thin-fin type, $t_{1}=A\cos\left(2\pi y/L\right)$ and $t_{2}=-A\cos\left(2\pi y/L\right)$ for big-source type, and $t_{1}=-A\cos\left(2\pi y/L\right)$ and $t_{2}=A\cos\left(2\pi y/L\right)$ for big-drain typ. Y-direction (or x-direction) means direction from upper source to bottom drain through gate length (or from left gate to right gate through fin width) in Figure 2.

**Figure 8 micromachines-12-01493-f008:**
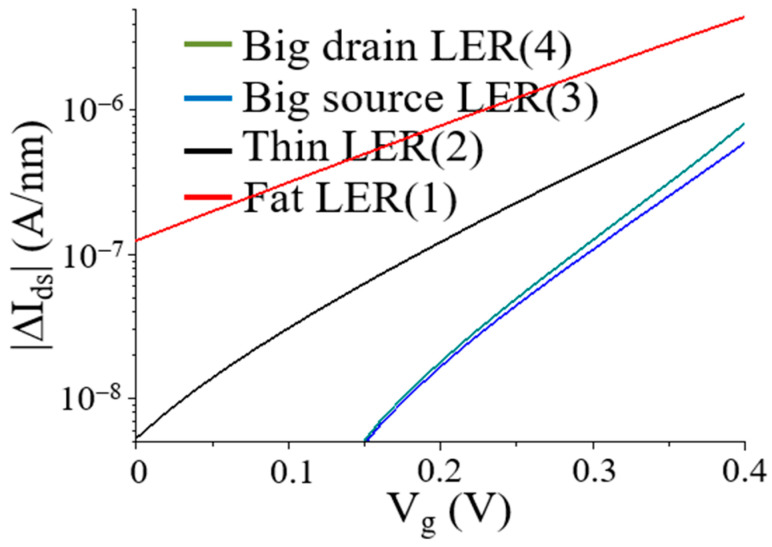
Analytical results of drain currents $\left|\Delta I_{ds}\right|$ with FWRs at gate length L = 5-nm due to gate voltages ($V_{g}$).

**Figure 9 micromachines-12-01493-f009:**
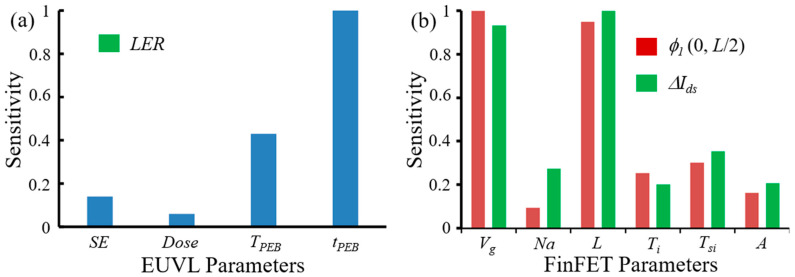
Parameter sensitivities: (**a**) *LER*; (**b**) electric potential ($\varphi_{1}$) and drain current ($\Delta I_{ds}$). Parameter sensitivities are normalized by the most sensitive parameter. EUVL parameters are secondary electron (*SE*), exposure dose (*Dose*), PEB temperature ($T_{PEB}$), and PEB time ($t_{PEB}$). FinFET parameters are gate voltage ($V_{g}$), doping concentration ($N_{a}$), channel length (L), oxide thickness ($T_{i}$), fin width ($T_{si}$), and FWR amplitude (A).

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
