# Peer review of "Line-Edge Roughness from Extreme Ultraviolet Lithography to Fin-Field-Effect-Transistor: Computational Study"

_micromachines, 2021, doi:10.3390/mi12121493_

Round 1

Reviewer 1 Report

Before July, 2021, TSMC announced the 3-nm manufacturing technology was still used with FinFET structure, not multi-nano-sheet device structure. Thus, I personally recognize your contribution in the deep nano-node process is still valuable. Your introduction part in description is good.

For writing concerns, some grammars must be corrected and some expressions are not clear. If possible, you need to check the full article or ask for the assistance from a native English speaker.

  1. In Figure 1, what are SE and Io? Is “SE” secondary electron?
  2. Couple of redundant words must be removed in conclusion part, such as (CD) and (EUVL). They are shown in the previous paragraphs.

        For the technical concerns, please comment them:

  1. In Fig. 5, the relationship between in Equation (9) and Fig. 5 with ti assumed as a rectangular shape must be illustrated more. In the real case, the photo diffraction will cause the rounding effect. The corner shape is changed. Thus, the ideal integration will be distorted. How do you confirm this error?
  2. Furthermore, the thickness of gate dielectric at the corner has the thinning effect. How do you overcome this difference between the ideal assumption and the real fabrication?

3. In the computational study with Monte Carlo simulator at the 5-nm node process, the consequences are too ideal to demonstrate the real circumstances. Please refer some published literature or documents to prove your simulation results. 

Author Response

Thank you for review. 

I include an answer sheet in attachment. 

Reviewer 2 Report

The paper describes a computational study of the Line-Edge Roughness from EUVL to Fin-FET using simulation. A series of analyses and calculations is demonstrated. However, some critical issues need to be fixed:

  1. I suggest using more references from updated published journal papers and not the proceeding or Ph.D. thesis work.
  2. In Figure 4b, the color legend is different from the curve lines color presented in the graph.
  3. In Figure 9a, the LER legend and the color of the bars are not associated. Please fix them.
  4. The manuscript should be sequentially written in Introduction, Methods, Results and Discussion, and Conclusion for better understanding for the readers.
  5. The methodology and parameters used in the Monte Carlo simulation should be mentioned in detail.
  6. Please explain in the manuscript how the Taguchi method was performed and how the parameters of EUVL and Fin-FET were determined in regards to sensitivity.
  7. Please rewrite the conclusion in a more concise and straightforward manner related to the most important findings. In addition, please emphasize the possibly suggested solution for LER impacts in Fin-FET.

Author Response

(The authors gave the same response as above.)

Round 2

Reviewer 1 Report

Your revised article is good, but the revised parts should be highlighted with color or color background. It's not easy to know you have changed what I concerned.